# Myths and Facts about Food Intolerance: A Narrative Review

**DOI:** 10.3390/nu15234969

**Published:** 2023-11-30

**Authors:** Fabiana Zingone, Luisa Bertin, Daria Maniero, Michela Palo, Greta Lorenzon, Brigida Barberio, Carolina Ciacci, Edoardo Vincenzo Savarino

**Affiliations:** 1Department of Surgery, Oncology and Gastroenterology, University of Padua, 35124 Padua, Italy; luisa.bertin.1@studenti.unipd.it (L.B.); daria.maniero@unipd.it (D.M.); michela.palo@unipd.it (M.P.); greta.lorenzon@unipd.it (G.L.); edoardo.savarino@unipd.it (E.V.S.); 2Gastroenterology Unit, Azienda Ospedale—Università Padova, 35128 Padua, Italy; brigida.barberio@unipd.it; 3Department of Medicine, Surgery and Dentistry, Scuola Medica Salernitana, University of Salerno, 84081 Salerno, Italy; cciacci@unisa.it

**Keywords:** FODMAP diet, food intolerance, fructose intolerance, sucrase-isomaltase complex, wheat hypersensitivity, lactose intolerance

## Abstract

Most adverse reactions to food are patient self-reported and not based on validated tests but nevertheless lead to dietary restrictions, with patients believing that these restrictions will improve their symptoms and quality of life. We aimed to clarify the myths and reality of common food intolerances, giving clinicians a guide on diagnosing and treating these cases. We performed a narrative review of the latest evidence on the widespread food intolerances reported by our patients, giving indications on the clinical presentations, possible tests, and dietary suggestions, and underlining the myths and reality. While lactose intolerance and hereditary fructose intolerance are based on well-defined mechanisms and have validated diagnostic tests, non-coeliac gluten sensitivity and fermentable oligosaccharide, disaccharide, monosaccharide, and polyol (FODMAP) intolerance are mainly based on patients’ reports. Others, like non-hereditary fructose, sorbitol, and histamine intolerance, still need more evidence and often cause unnecessary dietary restrictions. Finally, the main outcome of the present review is that the medical community should work to reduce the spread of unvalidated tests, the leading cause of the problematic management of our patients.

## 1. Introduction

In the last twenty years, cases of adverse reactions to food have significantly increased, with up to 20–35% of the Western population reporting symptoms after the consumption of different types of food [1,2,3,4,5]. However, food allergy or intolerance are well documented only among about 3.6% of the population [6]. These unspecified reactions often cause long-term dietary restrictions, leading to a worse quality of life, eating disorders, and dysbiosis [7]. Most of these food reactions are not attributed to allergic processes but are related to food intolerances, pharmacologic reactions, and toxic reactions [7]. Nevertheless, it is important to bear in mind that most of patients with functional GI disorders (FGIDs) report symptoms that they perceive as food-related. In some cases, a dietary restriction improves their symptoms [8]. Clinicians must maintain a high level of suspicion when evaluating a patient with possible adverse reactions to food to determine if a food allergy is at fault, if the reaction is due to one of the several other non-immune-mediated reactions, or caused by a nocebo effect (Figure 1) [7].

Earlier research identified connections between self-reported food intolerances and psychological symptoms, as well as their impact on the quality of life (QOL). Bohn et al. determined that food intolerances were linked to a notable burden of symptoms and a decline in QOL [9]. In a recent study by Claire L. Jansson-Knodell et al., the findings further supported associations between food intolerance and conditions such as anxiety and depression, reinforcing the connection with a diminished health-related QOL [10]. Moreover, patients who report adverse food reactions often resort to therapeutic interventions based on self-diagnosis without seeking medical or dietary consultation [11,12]. In this narrative review, we conducted a comprehensive revision of the available literature on the most common food intolerances with the aim of clarifying the myths and reality and giving clinicians a guide on how to manage these cases.

## 2. Non-Coeliac Gluten/Wheat Sensitivity—NCGWS

One specific category of food intolerance is associated with the consumption of gluten, giving rise to the so-called gluten-related disorders which are used to describe all conditions related to gluten such as coeliac disease (CeD), wheat allergy (WA), and gluten ataxia (GA) [13]. Coeliac disease and wheat allergies are well-known pathological conditions that provoke a distinct immune response.. CeD involves the production of autoantibodies primarily targeting the gastrointestinal tract, while wheat allergies trigger an IgE-mediated reaction [14]. However, a subgroup of individuals experience intestinal and extra-intestinal symptoms after consuming gluten but do not exhibit coeliac-specific antibodies or allergic biomarkers [14]. These individuals are categorized as having non-coeliac gluten sensitivity (NCGS), a condition first described in the 1980s when a 43-year-old patient presented diarrhea but lacked biomarkers for CeD. Remarkably, this patient demonstrated significant improvement after adopting a gluten-free diet (GFD) [15].

### 2.1. Epidemiology and Diagnosis

NCGS has been rediscovered in the 21st century due to its increasing incidence. It is estimated that the prevalence of NCGS ranges from 0.6% to 10% of the total population, with a higher predominance in females [16]. Currently, there are no specific tests or biomarkers available for diagnosing this condition. The only reliable method to identify people with gluten hypersensitivity is through an exclusion diagnosis, based on the Salerno criteria [17].

In 2014, the 3rd International Expert Meeting on Gluten-Related Disorders established the diagnostic criteria for NCGS [17]:Persistent intestinal and extra-intestinal symptoms while on a gluten-containing diet.Exclusion of CD through negative serology and absence of villous atrophy.Exclusion of wheat allergy through negative prick test and IgE levels.Improvement of symptoms after initiating a GFD for six weeks.Gluten challenge using a double-blind randomized placebo-controlled trial, which should result in a recurrence of symptoms with gluten intake but not with a placebo (at least a 30% reduction in one of the characteristic symptoms for 50% of the observation time).

Important factors contributing to the misdiagnosis of NCGS are firstly not testing for typical autoantibodies and more rarely seronegative coeliac disease.

A growing problem is the self-diagnosis of NCGS. Indeed, many individuals eliminate gluten-containing foods from their diet without undergoing medical investigations for specific gluten-related disorders. It has been estimated that this self-diagnosis carries a risk of 2–42% of undiagnosed CeD, leading to incorrect treatment [18]. Instead, in cases of suspicion of seronegative coeliac disease, it is crucial to investigate the presence of duodenal damage while on a gluten-containing diet [19]. Genetic predisposition is not useful for diagnosing NCGS, as NCGS and CeD patients share 50% of the HLA DQ2/8 haplotypes [20]. Additionally, the misconception that a GFD is a healthy lifestyle may lead patients to avoid gluten-containing foods, regardless of the side effects such as the higher fat content, the loss of dietary fiber, and deficiencies in dietary minerals and vitamins [21].

### 2.2. Clinical Features and Pathogenesis

NCGS is characterized by intestinal and extra-intestinal symptoms that occur following gluten consumption [22]. The typical intestinal symptoms include bloating, abdominal or epigastric pain, diarrhea, and nausea. Extra-intestinal manifestations may include fatigue, headache, a feeling of mental fogginess, and depression [22]. Often, many NCGS patients have a family history of CeD or food allergies and are also associated with other autoimmune diseases such as Hashimoto’s thyroiditis and diabetes mellitus [23]. Symptoms usually appear within hours to days after gluten ingestion and disappear when gluten is removed from the diet [22]. Patients who do not respond to a GFD should be investigated for other potential causes, such as irritable bowel syndrome (IBS)-like symptoms, given that, as reported above, an overlap between IBS with NCGS is quite common [18].

Regarding pathogenesis, there is evidence suggesting that the innate immune response plays a primary role rather than the adaptive immune response [17]. Toll-like receptors (TLR 2-4) on the epithelial level become activated in response to pathogenic microorganisms [17]. Furthermore, the involvement of the intestinal microbiota has been described, with an increase in pathogenic *Bacteroidetes* and a decrease in saprophytic bacteria (*Firmicutes*) in patients with NCGS [22]. This dysbiosis may contribute to the bloating experienced by patients, potentially due to increased fermentation. At the molecular level, an increase in Claudin-4, an integral component of tight junctions responsible for paracellular permeability, has been observed [24]. Therefore, similar to CeD, the “leaky gut” hypothesis is proposed for NCGS, suggesting an impaired intestinal barrier [25].

From a histological point of view, there are not many differences compared to healthy individuals, as a normal villous architecture is maintained at the duodenal level [26]. A cut-off value of intraepithelial lymphocytes (IELs) ≤25/100 enterocytes is used to define normal duodenal histology, corresponding to Marsh 0 [26]. However, patients classified as NCGS show an increased count of duodenal IELs (>25/100 enterocytes), corresponding to Marsh I lesions. A recent multicenter study observed that NCGS duodenal mucosa exhibits distinctive changes consistent with an intestinal response to luminal antigens, even at the Marsh 0 stage of villus architecture [27]. Recent studies have also found the presence of eosinophils at the lamina propria level in NCGS patients, suggesting a condition that may be closer to a food allergy [28,29,30].

The serum zonulin dosing for NCGS is still a topic of ongoing research and is not yet established as a standard diagnostic or treatment method [31]. Zonulin is a protein that plays a role in regulating the permeability of the intestinal barrier [32]. It has been suggested that in NCGS in HLA-DQ2/8-positive individuals, increased zonulin lowers with a GFD and that it could be used in the differential diagnosis with IBS-D [31].

### 2.3. Gluten or Not Only Gluten, That Is the Question

NCGS terminology remains a topic of ongoing debate. Many authors prefer the term “non-coeliac wheat sensitivity” (NCWS) due to recent studies highlighting that other proteins in wheat can trigger the symptoms of intolerance [33]. Wheat, barley, and rye are gluten-containing grains, and wheat, in particular, is one of the most common and widespread cereals cultivated globally [34]. Wheat has the ability to grow in different environments and its high nutritional value makes it a fundamental food worldwide. Wheat contains various proteins that can be classified as structural, functional, or storage proteins, depending on their function. Among these, 80% are storage proteins, including gliadins and glutenins, which contribute to the formation of gluten. These storage proteins are rich in proline amino acids, which makes them resistant to intestinal proteolytic enzymes [35]. The high proline content leads to the production of immunogenic peptides, which can trigger inflammatory and oxidative stress responses [36].

However, the exact role of gluten as a culprit in intolerance is not completely understood. Wheat contains 2–4% of amylase-trypsin inhibitors (ATIs). These proteins play a role in the plant’s natural defense against insects and parasites, but they also possess enzymatic activities that degrade nutrients, such as α-amylase involved in starch breakdown and trypsin involved in protein degradation [37].

Experimental evidence has shown that ATIs can exacerbate the immune response in the intestine and increase inflammation, particularly in individuals with the HLA/DQ8 haplotype [38].

Another group of compounds comprises fermentable oligosaccharides, disaccharides, monosaccharides, and polyols (FODMAPs), which are short-chain carbohydrates, including fructans and galacto-oligosaccharides (GOS), and are found in various foods, including wheat. They are slowly absorbed in the small intestine and undergo rapid fermentation by gut bacteria, resulting in gas production and bloating [39,40]. FODMAPs have been implicated in triggering symptoms in various gastrointestinal disorders, including NCGS [41]. Biesiekierski JR. observed for the first time the positive effect of a low FODMAP diet in a group of self-reported NCGS [42].

Skodje et al. conducted a double-blind crossover challenge to examine the impact of gluten and fructans in individuals with self-reported NCGS. The results revealed that, in this group, it is fructans, rather than gluten, that appear to trigger more significant gastrointestinal symptoms [43].

An observational study found that NCGS patients eat different foods than healthy individuals; patients consume lower levels of proteins, carbohydrates, fiber, and polyunsaturated fatty acids, suggesting that dietary advice is often needed to avoid nutritional imbalance [44].

### 2.4. Conclusions

To summarize, NCG/WS is a condition that is not yet fully understood, presenting difficulties in terms of diagnosis due to its different range of symptoms, which can overlap with IBS. Currently, exclusion diagnosis is the primary method used to diagnose NCG/WS. It is important to rule out CeD and WA through serology tests after a proper reintroduction of gluten-containing foods if they had been previously eliminated [17]. It is important to underline that there is evidence that gluten is not the only trigger of this condition. Wheat contains other proteins such as ATIs and FODMAPs, which contribute to the typical manifestations of NCG/WS [38,39]. Attention should be focused on individuals who are already on a GFD, without a formal medical diagnosis suggesting this diet [18]. Some individuals eliminate gluten from their diet based on the belief that it is the cause of their symptoms, due to a family history of food allergies, or as a preventive measure against CeD [45].

## 3. FODMAP Intolerance

FODMAPs are short-chain carbohydrates that include lactose, fructose when in excess of glucose, sugar polyols (sorbitol and mannitol), fructans, and GOS (stachyose and raffinose) naturally present in a large number of foods like fruits, vegetables, cereals, dairy products, and sweeteners [39,40] (Figure 2). High-FODMAP foods are those that contain more than 4 g of lactose, more than 0.3 g of mannitol, sorbitol, galacto-oligosaccharides, or fructans. So, it is possible to categorize foods by considering their FODMAP amount [46].

FODMAPs are poorly absorbed and fermented by intestinal bacteria. In this way, the consumption of high amounts of FODMAPs leads to the excessive production of short-chain fatty acids (SCFAs) and of a large amount of gas, including carbon dioxide, hydrogen, and methane, that are responsible for luminal distention and luminal water retention [47]. Thus, gastrointestinal (GI) symptoms, such as bloating, abdominal pain, flatulence, and diarrhea, occur in susceptible individuals, particularly in patients affected by IBS [47,48].

### 3.1. Uses of Low-FODMAP Diets in Clinical Practice

IBS is a functional GI disorder that affects up to 20% of people worldwide [49]. Gastrointestinal symptoms are influenced by different factors, such as the psycho-social sphere, physiological functioning, and their interaction (gut–brain axis) [50]. IBS is characterized by anomalous visceral motility and sensibility, and abnormalities in immune function and microbiota composition; thus, it is associated with various GI symptoms and an impaired quality of life (QoL) [51]. The Rome IV diagnostic criteria permit the division of IBS patients into three categories, depending on their symptoms: IBS with diarrhea (IBS-D), IBS with constipation (IBS-C), and IBS with mixed bowel habits (IBS-M) [49].

Nowadays, the pathogenesis of IBS is still not completely understood, but different studies have demonstrated that diet plays an important role in symptom management [46]. The most evidence-based diet is the low-FODMAP diet (LFD), which was found to be effective in IBS treatment [49,52]. In particular, the randomized control study by Halmos et al. compared the effects of the LFD and the Australian diet on IBS patients, and showed that the LFD resulted in being effective in reducing functional GI symptoms, measured with the visual analog scale [53]. Moreover, the recent review by Morariu et al., which included seven studies, confirmed the positive effects of the LFD, showing that the IBS severity scoring system (IBS-SSS) decreased significantly after the LFD. Likewise, QoL improved compared with patients following a standard diet [46]. Finally, the efficacy of the LFD seems to be higher in IBS-D and IBS-M patients, compared to IBS-C ones, as evidenced by the randomized controlled trial by Algera et al. This latter study also demonstrated that an LFD is more effective than a moderate FODMAP diet in IBS treatment [54].

### 3.2. The Low-FODMAP Diet Approach

Focusing on the characteristics of the LFD, it can be applied with the “top-down” approach, divided into three steps: restriction, followed by reintroduction and personalization for long-term maintenance (Figure 3).

Phase one requires the exclusion of all high-FODMAP foods from the diet, usually for a period of 4–6 weeks [55]. This is the most critical phase, in which it is important to explain to the patient the role of FODMAPs in the occurrence of GI symptoms and to provide counseling regarding the foods with the highest FODMAP contents and how to avoid them. Moreover, a clarification about the diet, its timeline, and what to expect from it is necessary to implement diet adherence [56].

Phase two aims to reintroduce FODMAPs and to assess tolerance for each patient, to adjust the diet for long-term maintenance [55]. In this phase, it is also important to evaluate diet compliance and clinical response; a personalized reintroduction of foods, considering 3 days for each one, permits the identification of patients’ specific triggers of symptoms [57].

Phase three involves the development of a long-term diet, customized to align with the patient’s FODMAP tolerance [55]. The key point is to structure a flexible diet, in order to maintain both variety and nutritional adequacy, and also control IBS symptoms. In fact, a high percentage of patients, following a properly “modified LFD”, continue to benefit from it after 6–18 months [57,58].

An alternative approach, defined as “bottom-up”, is possible for patients who cannot deal with dietary restrictions [55]. It consists of a gentle LFD, with the exclusion of one or two subgroups of high-FODMAP foods from the diet and response evaluation. In case of symptom persistence, further restrictions need to be applied. Data regarding this approach are limited, so further research is necessary to understand its effectiveness [55,58].

### 3.3. The LFD Consequences

#### 3.3.1. Nutritional Consequences

In recent years, different authors have investigated the nutritional adequacy of the LFD. The main reason for nutritional deficiencies is the absence of appropriate dietary counseling and a self-restricted diet. Studies on dietary intake are discordant. Fiber deficiencies are the most frequent, due to the reduction in carbohydrate intake; moreover, calcium intake was lower when an excessive dairy product exclusion was applied. Considering vitamin consumption, the risk of deficiencies is linked to a strict reduction in vegetables and fruits in the diet [56,59]. Finally, lower energy consumption with LFD may lead to weight loss [59].

It is important to underline that data regarding the long-term effects of LFD are lacking, but if patients are properly monitored by health professionals during the course of diet, the risks of nutritional deficiencies are very low [46,57].

#### 3.3.2. Constipation

Another potential limit of the LFD is the low consumption of fibers, which may worsen constipation, especially in IBS-C patients [60]. Bellini et al. reported that fiber deficiency is quite common in patients following an LFD, while a review by Sultan et al. evidenced that studies previously conducted are discordant [57,59]. The recent randomized controlled crossover trial conducted by So et al. demonstrated that fiber supplementation during the LFD did not modify patients’ bowel movement perception, but it normalized water stool content and colonic transit [61].

As reported before for nutritional risks, a proper nutritional follow-up and patient education are indicated to improve diet management and fiber intake [57].

#### 3.3.3. Eating Disorders

The restrictions applied during the first step of the LFD may negatively impact the emotional status of patients. In fact, a recent study conducted by Rei et al. evidenced that IBS patients following an LFD had a lower QoL in relation to food [62]. Moreover, the LFD, the anxiety over worsening IBS symptoms, and the consequent dietary restrictions seem to correlate with the onset of eating disorders [57]. In particular, avoidant restrictive food intake disorder and orthorexia nervosa are the most common eating disorders associated with LFD [57,59]. For these reasons, a screening for eating disorder assessment is recommended before starting the LFD [63].

#### 3.3.4. Microbiota

Nowadays it is known that IBS patients present an altered gut microbiota compared with healthy subjects [64]. Thus, recent studies focused on detecting microbiota changes after the LFD, particularly variations in terms of composition and functioning [65]. A meta-analysis by So et al. reported only the reduction in *Bifidobacteria*, and similar results were reported by van Lanen et al. in their metanalysis, while the microbiota function did not change [65,66]. Recent studies also tried to identify potential microbiota biomarkers to predict LFD response. There are two different IBS microbiota subtypes, linked to different responses to the LFD [64,66]. To conclude, further research in this new field is necessary, with the hope of identifying IBS patients who can benefit from the LFD and improve their management. 

## 4. Lactose Intolerance

Lactose intolerance (LI) presents as a clinical syndrome characterized by specific signs and symptoms, including abdominal pain, bloating, and diarrhea, triggered by consuming lactose in individuals with lactose malabsorption (LM). Normally, lactose, a disaccharide, is broken down into glucose and galactose by the lactase enzyme, found in the small intestine’s brush border. Lactase deficiency is common in healthy individuals, resulting in LM when they consume milk or lactose-containing foods. LM can have primary or secondary causes and is a necessary precondition for LI. However, it is not sufficient, since LI is not always present in cases of LM.

Individuals reporting LI symptoms without formal testing are considered to have self-reported lactose intolerance, while those testing negative for LM are classified as having functional lactose intolerance [67]. However, it must be noted that there is no correspondence between symptoms and a positive test for LI [68].

The primary cause of LM is lactase non-persistence (LNP), where intestinal lactase expression decreases during the first two decades of life. Recent findings suggest that LNP is the ancestral form, following normal Mendelian inheritance, while lactase persistence (LP) arises from specific mutations [69]. LNP’s prevalence varies based on regional genetic heritage, with higher rates in South American, Asian, and African descent populations, and lower rates in individuals of northern European or northwestern Indian origin [70].

The global prevalence of LM is around 68%, with higher rates reported in genetic tests compared to hydrogen breath tests (H_2_BTs). LM is least prevalent in Nordic countries and highest in Korean and Han Chinese populations [71]. The prevalence of LI is currently unknown due to the complexities of testing and diagnosis [67].

### 4.1. Causes of Lactase Deficiency

There are four leading causes.

Primary Lactase Deficiency or lactase non-persistence

Lactase deficiency is characterized by a gradual decline in lactase enzyme activity as individuals age. The decline begins during infancy, and symptoms typically manifest in adolescence or early adulthood [69]. In the Caucasian population, LP results from a gain-of-function mutation (LCT-13′910:C→T, referred to as ‘T’ for tolerance) on chromosome 2 [69] and this genetic mutation is considered a dominant genotype (LCT-13′910:CT and LCT-13′910:TT), while only individuals with two LCT-13′910:C (LCT-13′910:CC) alleles are classified as LNP [69]. Importantly, LNP is not a disease but rather a genetic wild type [69]. Both LP and LNP are common phenotypes observed in healthy individuals [69].

Secondary Lactase Deficiency

Secondary lactase deficiency arises from damage to the intestinal epithelium and can occur in various conditions such as gastroenteritis, chemotherapy, antibiotic usage, celiac disease, inflammatory bowel disease, AIDS, malnutrition, or conditions that reduce the absorption surface, like short bowel syndrome [72]. The reduction in lactase activity is temporary and reversible, and it typically improves once the underlying intestinal damage is treated or resolved [73].

Congenital Lactase Deficiency

Congenital lactase deficiency is an extremely uncommon pediatric disorder that results in severe symptoms and failure to thrive in infants [74]. The condition arises from a genetic inheritance pattern known as autosomal recessive, leading to reduced or absent lactase enzyme activity from birth [75].

Developmental Lactase Deficiency

Developmental lactase deficiency is observed in premature infants born between 28 to 37 weeks of gestation. In these cases, the infant’s underdeveloped intestine leads to an inability to break down lactose. However, this condition typically improves with age as the intestine matures and with feeding, especially breastfeeding [76].

### 4.2. Clinical Characteristics

LM, whether due to primary or secondary lactase deficiency, leads to undigested lactose interacting with the intestinal microbiota [77]. Bacterial fermentation of lactose results in the production of short-chain fatty acids (acetate, propionate, and butyrate) and gas (hydrogen, carbon dioxide, and sometimes methane) [78]. Diarrhea occurs when the amount of lactose exceeds the capacity of the colonic microbiota for fermentation or when the load of short-chain fatty acids exceeds the colon’s capacity for resorption [71]. The osmotic trapping of water further increases the osmotic load in the colon, amplifying the effect [79]. However, individuals may suffer from lactase deficiency and not have symptoms. In cases of clinical manifestations, symptoms include bloating, abdominal pain, flatulence, diarrhea, and sometimes nausea [80]. The severity of symptoms may vary, and most individuals may tolerate relatively small amounts of lactose without discomfort [80]. The likelihood of developing symptoms after lactose ingestion is influenced by various factors [67]. Extrinsic factors include the quantity of lactose consumed and whether dairy products are consumed alongside other foods affecting intestinal transit and lactose delivery rate to the colon [81]. Intrinsic factors involve the expression of lactase in the small intestine, history of GI disorders or abdominal surgery, intestinal microbiome composition, visceral hypersensitivity, anxiety, and the presence of disorders of gut–brain interaction (DGBIs) [82].

The intestinal microbiota usually adapts itself to facilitate dairy product intake, leading to reduced lactose intolerance symptoms with regular lactose consumption and an increase in healthy components of the gut microbiome, such as Bifidobacteria and Lactobacilli [81]. Additionally, interactions between human genes and the microbiota, such as the LCT-13′910:C/T SNP’s association with the abundance of Bifidobacterium, have been observed [83].

The severity of symptoms induced by lactose is notably increased in IBS patients, especially at the lower to moderate lactose doses found in a normal diet [84]. A meta-analysis conducted in 2018 by Varjú et al. confirmed that self-reported LI but not LM is more frequent in patients with IBS than in healthy controls, further underlining the differences between LI and LM [85].

### 4.3. Diagnosis

Various diagnostic methods can be utilized to detect LM, including genetic testing, enzymatic assays, and breath tests [86,87,88]. Among these methods, measuring lactase enzyme activity in small bowel biopsies is considered the most specific [89]. However, the lactose breath hydrogen test is a preferred non-invasive technique for evaluating lactose digestion and related symptoms [90] Appendix A.

These diagnostic tests have a major limitation: as previously described, LM is commonly found in healthy individuals, i.e., individuals not reporting gastrointestinal symptoms after lactose ingestion, and thus a positive test result does not necessarily confirm that symptoms are caused by LM [91]. This limitation has been addressed by utilizing a standardized symptom questionnaire during hydrogen breath testing or by blinded testing [92]. Blinded testing could provide valuable insights in such cases, particularly due to the low correlation between self-reported symptoms of LI and objective findings in tests for LM [86,93]. This lack of correlation is even more significant in patients with IBS than in healthy individuals [93]. Most importantly, a hydrogen lactose breath test is generally performed using the standard dosage of 1 mg/kg (usually from 20 to 25 g of lactose), in a single dose, corresponding to the quantity contained in more than 500 mL of milk [86]. It is known that only about 50% of the enzyme is required for a breakdown of lactose, and thus the standard hydrogen breath test does not indicate the daily amount of lactose that a person with proven LI could tolerate, especially in separate portions throughout the day [94]. A ‘blinded multiple dose challenge’ would help to understand lactose digestion and identify the amount of lactose that individuals could ‘safely’ consume [94].

### 4.4. Treatment

Treatment options for LI encompass a range of approaches, including adopting a low-lactose diet, using oral lactase enzyme replacements, employing prebiotics to stimulate bacterial lactase production in the colon, and potentially employing prebiotics to modify the colonic microbiota [95]. It is essential to differentiate between LM and LI since a lactose-restricted diet is only necessary for patients with intolerance [96].

#### 4.4.1. Diet

For most individuals with LI, reducing lactose intake rather than completely excluding it from the diet is sufficient [96]. Consuming smaller lactose doses (e.g., 12 g of lactose, equivalent to 200–250 mL of milk) alongside other foods is often well-tolerated and may have benefits over a strict lactose-free diet [97].

To limit the possibility of having symptoms, one solution may be taking lactose with other foods to slow gastric emptying and small intestinal transit, allowing more time for lactose to be broken down and absorbed, reducing the likelihood of symptoms. It would be useful to encourage the consumption of aged cheeses which, unlike fresh ones, contain little to no lactose. In fact, during the ripening process, the bacteria consume all the lactose present.

To ensure the intake of the substances contained in dairy products without causing abdominal discomfort due to lactose, the production of lactose-free foods was initiated. From a nutritional point of view, they are comparable to classic dairy products, with the difference that they do not contain lactose [98]. Dairy products devoid of lactose, supplemented with added lactase enzyme, are readily accessible and generally deemed safe. Ongoing advancements in technology are continuously improving the nutritional value, functionality, sensory appeal, and quality of lactose-free dairy products [99]. These developments aim to offer lactose-intolerant individuals more diverse and palatable options while ensuring optimal nutritional intake and overall well-being [99].

#### 4.4.2. Oral Lactase Enzyme Replacement

Lactase supplementation in the form of tablets has shown improvements in both lactose digestion, leading to reduced hydrogen (H_2_) production, and symptom relief [100,101].

Enzymatic integration using exogenous lactase derived from non-human sources presents a viable option. This lactase can be obtained from yeast, such as *Kluyveromyces lactis*, or fungi, including *Aspergillus oryzae* and *Aspergillus niger* [81]. However, it is worth noting that isolated instances of allergic reactions have been reported [102].

Ibba et al. conducted a study to evaluate the efficacy of exogenous lactase in lactose-intolerant subjects. The enzymatic compound exploited by these authors was beta-galactosidase, obtained from the fermentation of *Aspergillus oryzae*. A reduction in hydrogen excretion, as measured by H_2_BT, was achieved in 40% of patients. On the other hand, in the remaining 60% of them, the amount of hydrogen excreted did not change and the effects on symptoms were modest, as only about 18% of patients experienced a reduction in symptoms [101].

#### 4.4.3. Probiotics

Another approach is the use of probiotics, such as *Lactobacillus* spp., *Bifidobacterium longum*, or *Bifidobacterium animalis*, which produce lactase in the gut.

A systematic review published in 2022 included a total of three studies using the probiotics *Bifidobacterium bifidum* 900791, *Limosilactobacillus reuteri* DSM 17938 (*Lactobacillus reuteri*), and *Lactobacillus acidophilus* DDS-1 comprising a total of 117 subjects [103]. The results showed that only *Limosilactobacillus reuteri* DSM 17938 showed significant improvement in symptoms and reduction in expired hydrogen, while *Lactobacillus acidophilus* DDS-1 showed significant improvement in LI symptoms [103].

Probiotics are often added to dairy products, both as fermenting agents and as food additives; however, although their efficacy has been evaluated by some studies, there is not enough evidence to suggest them as a therapeutic option [104,105]. A recent meta-analysis, which included 12 studies aimed at investigating the efficacy of probiotics in patients with LI including a total sample size of 263 patients, found that probiotic administration alleviated the symptoms of LI [106].

#### 4.4.4. Prebiotics

Improved lactose tolerance by manipulating the colonic microbiota could also be achieved by ingestion of prebiotics [107].

A recent systematic review included two studies in which the efficacy of short-chain GOS (RP-G28) was studied in a total of 462 subjects. The authors found that GOS (RP-G28) showed improvement in LI symptoms during the treatment phase and up to 30 days after its cessation [103].

A randomized placebo-controlled study in 377 LI patients reported that regular ingestion of short-chain GOS (RP-G28) found significant improvements in global assessments compared to placebo and significant increases in five *Bifidobacterium* taxa [107].

## 5. Hereditary Fructose Intolerance

Hereditary fructose intolerance (HFI) is a rare autosomal recessive disorder caused by a mutation in the aldolase B enzyme located on chromosome 9q22.3 that can cause, after fructose ingestion, significant gastrointestinal symptoms and potentially lead to long-term organ damage, particularly renal and hepatic.

HFI is characterized by the inability to metabolize fructose properly, leading to various metabolic disturbances and clinical symptoms. Fructose, found in honey, fruits, and many vegetables, is absorbed from the intestine through glucose transport proteins (GLUT) 5 and 2. Enzymes such as fructokinase, aldolase B, and triokinase are responsible for fructose metabolism in the liver, kidney, and small intestine. Deficiency in aldolase B results in the abnormal accumulation of fructose-1-phosphate (F-1P), depleting intracellular inorganic phosphate and adenosine triphosphate. This leads to impaired protein synthesis, adenosine monophosphate consumption, and inhibition of glycogenolysis and gluconeogenesis, causing hypoglycemia [108].

### 5.1. Clinical Characteristics

HFI symptoms manifest when individuals with this condition are exposed to dietary fructose directly or indirectly through sucrose or sorbitol. The disease is typically diagnosed in infants but can also present later in childhood or adulthood due to voluntary strict dietary restrictions [108].

Individuals with fructose intolerance often experience GI symptoms such as abdominal pain, bloating, diarrhea, and nausea after consuming foods or beverages high in fructose [109]. These symptoms can range from mild to severe discomfort and can significantly impact an individual’s quality of life [108]. Symptoms and their severity depend on fructose dosage, patient age, concomitant diseases, and residual enzymatic activity of aldolase B and are non-specific, making it difficult to suspect HFI based on symptoms alone [110]. Common clinical findings are nausea, vomiting, abdominal distress, and growth restriction/failure to thrive [108].

The clinical manifestations can also appear following administration of either of the two sucrose-containing rotavirus vaccines, Rotarix^®^ and RotaTeq^®^ [111]. However, untreated HFI is also characterized by metabolic disturbances (hypoglycemia, lactic acidosis, hypophosphatemia, hyperuricemia, hypermagnesemia, and hyperalaninemia). Chronic ingestion of fructose can lead to hepatic or renal injury and growth disturbance [108]. Liver manifestations include elevated liver enzymes, steatohepatitis, cirrhosis, and occasionally acute liver failure, while renal involvement often presents as proximal renal tubular acidosis and may lead to chronic renal insufficiency [112,113]. Some patients with residual aldolase B activity may have subtle symptoms and an aversion to sweets [114].

### 5.2. Diagnosis

Diagnostic tests for HFI include a screening test involving the association of Benedict’s test and glucose dipstick test to detect fructose in the urine and elevated serum carbohydrate-deficient transferrin (CDT) levels in patients with suspected HFI based on metabolic disturbances and/or clinical findings [115]. Family history may be present, but it is not necessary for the diagnosis as the disease has an autosomal recessive inheritance [108]. If the ALDOB pathogenic variants have been identified in an affected family member, carrier testing for at-risk relatives, and prenatal testing for a pregnancy at increased risk can be undertaken [116]. It is suggested to test siblings even before symptoms occur [117]. Genetic testing requires no further confirmation as it is highly specific, sensitive, and less invasive than measuring aldolase B activity in liver biopsy specimens [118]. Molecular testing aims to find biallelic pathogenic (or likely pathogenic) variants in the ALDOB gene, while finding an ALDOB variant of uncertain significance does not allow the diagnosis to be made [116].

Alternatively, fructose-1-phosphate aldolase B enzyme assays and fructose assay enzyme panels on frozen liver tissue may be important options to establish the diagnosis in individuals with clinical and biochemical features of HFI in whom molecular genetic testing has failed to identify biallelic ALDOB pathogenic variants [114].

### 5.3. Treatment

The management of HFI involves strict avoidance of foods containing fructose, sucrose, and sorbitol (FSS) [109]. With proper diagnosis and adherence to a fructose-restricted diet, individuals with fructose intolerance can effectively manage their condition and improve their overall well-being [117].

In acute metabolic crises, patients require admission to an intensive care setting for intravenous glucose administration, treatment of metabolic acidosis, and supportive care [119,120,121]. It is strongly recommended to exercise special caution during hospitalizations to refrain from using intravenous fluids containing fructose, as well as avoiding fructose-containing infant formulas and pharmaceuticals [120].

Following a strict FSS-free diet, along with carbohydrate supplementation from sources such as glucose and corn starch, results in the rapid reversal of symptoms. Patients should avoid medications and vaccines containing sucrose such as the two live oral rotavirus vaccines, Rotarix^®^ and RotaTeq^®^. Given that reduced fruit and vegetable intake is a dietary requirement, daily supplementation with a “sugar-free” multivitamin is recommended to prevent micronutrient deficiencies, specifically water-soluble vitamins [108].

Long-term dietary compliance is essential and may require repetitive counseling, clear instructions, and continuous reinforcement to prevent breakthrough events. It has been suggested to periodically evaluate the liver and hepatic function, assess compliance, and enhance it by reinforcing indications. Some studies show that patients with HFI who adhere strictly to an FSS-free diet may have a good prognosis and normal lifespan [110,114], but data on the long-term outcomes of these patients are lacking. The main downside of a strict FSS is the development of nutritional deficiencies, such as vitamin C and B deficiencies [122].

## 6. Non-Hereditary Fructose Intolerance

The literature describes a non-hereditary fructose intolerance caused by an insufficient uptake of fructose into enterocytes relative to the amount of luminal fructose [123]. Fructose absorption capacity in the small intestine is much lower than glucose absorption capacity; glucose stimulates fructose absorption in a dose-dependent manner, and malabsorption occurs when fructose is present more than glucose [124]. Unabsorbed fructose then passes into the colon and is fermented in the same manner as lactose in patients who have LNP [124]. Diagnosing fructose malabsorption (FM) typically involves a combination of medical history assessment, symptom evaluation, and specific tests. A specific hydrogen breath test has been used for years to assess fructose malabsorption. The European guidelines on the indications, performance, and clinical impact of hydrogen and methane breath tests recommend that the dose of fructose in adults for the diagnosis of fructose malabsorption and intolerance should be 20–25 g [125]. However, the clinical utility of fructose HBT is debated. In fact, both the Rome Consensus Conference on ‘Methodology and indications of H2-breath testing in GI diseases’ and the ESPGHAN Position Paper on the Use of Breath Testing stated that a fructose breath test is not recommended in clinical practice [126,127]. Similarly, recent guidelines on chronic diarrhea do not recommend the use of carbohydrate breath tests in the diagnostic flow chart [128]. Given this diagnostic uncertainty, an elimination diet may be recommended, where high-fructose foods are removed from the diet for a specific period [122]. If symptoms improve during this time, and then return when fructose-containing foods are reintroduced, this can suggest fructose malabsorption. Since fructose is a FODMAP, a low-FODMAP diet is often recommended instead of a low-fructose one, particularly in patients suspected to have a concomitant functional disorder. Prior investigations have indicated that a substantial proportion, ranging from 35% to 73%, of individuals diagnosed with IBS or FGIDs exhibit characteristics consistent with FM [129]. Nevertheless, it is imperative to note that the absence of a universally acknowledged gold standard for FM diagnosis may introduce variability in the accuracy of these reported percentages [130]. Recently, a ‘Carbohydrate Perception Questionnaire’ was proposed as an instrument for the assessment of symptoms developed after carbohydrate ingestion, with excellent psychometric properties [92]. Xylose isomerase has been proposed as a potential treatment for fructose intolerance because of its ability to convert fructose into glucose. A double-blind, placebo-controlled study showed that oral administration of xylose isomerase was associated with a significant reduction in breath hydrogen after fructose ingestion, as well as a significant improvement in nausea and abdominal pain [131]. Currently, empirical therapy involves adopting a restricted diet and assessing the symptoms.

## 7. Saccharose Intolerance

Sucrose, or saccharose, consists of one glucose and one fructose molecule. The binding between these two molecules is broken by the membrane-bound enzyme sucrase-isomaltase [132]. Congenital sucrase-isomaltase deficiency (CSID) is a rare autosomal recessive condition with mutations of the sucrase-isomaltase gene on chromosome 3q25-26 [133]. Acquired forms of sucrase-isomaltase deficiency may be secondary to other chronic gastrointestinal conditions associated with intestinal villous atrophy, such as enteric infection, coeliac disease, Crohn’s disease, and other enteropathies affecting the small intestine. Functional sucrase-isomaltose genetic variants appear to be more common in patients with symptoms suggestive of IBS [132]. However, as reported above, recent guidelines do not recommend the use of carbohydrate tests in these patients [128].

The use of sacrosidase, an enzyme produced by Saccharomyces cerevisiae that hydrolyses sucrose, was suggested as a possible treatment of this intolerance since an old double-blind study revealed that this enzyme, administered along with food, significantly prevents symptoms of intolerance in patients on a sucrose-containing diet as compared with placebo [134].

## 8. Histamine Intolerance

The term histamine intolerance (HIT) was coined to draw a comparison with lactose intolerance [135,136]. HIT is regarded as a non-immunological condition believed to result from an imbalance between histamine uptake through the diet and a diminished capacity to metabolize ingested histamine, leading to an increased blood histamine concentration which may potentially cause adverse effects [137]. It is important, to distinguish histamine intolerance from histamine intoxication. Histamine intoxication arises from the ingestion of histamine-rich foods, with symptoms swiftly emerging, typically within minutes to a few hours post-meal [136], and is caused by higher levels of histamine, 100 mg in a mild case and above 1000 mg in a severe form [135,136]. It is characterized by occurring in outbreaks. The symptoms are intimately tied to the diverse physiological roles of histamine within the body, impacting the skin (resulting in effects like redness, rashes, hives, itching, swelling, and localized inflammation), the gastrointestinal system (manifesting as nausea, vomiting, and diarrhea), as well as affecting hemodynamic aspects (such as lowered blood pressure) and neurological functions (giving rise to symptoms like headaches, palpitations, and tingling sensations) [136].

### 8.1. Etiopathogenesis

Histamine, a biogenic amine, is synthesized endogenously from the amino acid histidine and plays a role in numerous physiological processes. It exerts predominantly local effects mediated by four receptors named H1, H2, H3, and H4. Histamine H1 receptors promote blood vessel dilation, airway constriction, and itching. H2 receptors regulate gastric acid secretion [138]. H3 receptors modulate the sleep–wake rhythm. H4 receptors influence the immune system. Histamine release and effects are tightly regulated at the cellular and local levels. Histamine is primarily stored in mast cells and basophils and serves as a major mediator of IgE and non-IgE-mediated immunological responses [138]. Histamine is metabolized through two pathways: methylation by histamine *N*-methyltransferase (HNMT), present in most body tissues, and oxidative degradation by diamine oxidase (DAO), which is a secretory enzyme mainly located in the small intestinal mucosa and kidneys [139]. Elevated histamine availability may result from various factors, including endogenous histamine overproduction due to allergies, mastocytosis, GI bleeding, or increased intake of histidine or histamine from food or alcohol [140]. However, current evidence regarding increased histamine plasma levels in patients with histamine intolerance is limited [141,142]. A recent proposal also suggests HIT can arise from an alteration in the gut microbiota with a greater abundance of histamine-secreting bacteria in the gut leading to its development [143].

Moreover, the main suspected cause of HIT is insufficient DAO activity or reduced levels thereof [137]. DAO activity that is compromised may result from either genetic inheritance or external factors, wherein specific single-nucleotide variations are linked to the lowered transcriptional activity of the DAO gene or a decrease in enzyme effectiveness [140]. This impairment may also be secondary to pathological or pharmacological factors [136,144,145,146]. Acquired histamine intolerance may be temporary and can be reversed by the discontinuation of the use of DAO-blocking medications, for example, acetylsalicylic acid or naproxen [147]. Several small bowel pathologies, affecting mucosal integrity, are known to result in impaired DAO activity, which correlates with the severity of the mucosal damage [144]. Thus, DAO activity has been proposed as a marker of integrity of the intestinal mucosa and has been linked to carbohydrate malabsorption and NCGS [148]. However, there is a lack of evidence supporting a direct association between enzyme deficiency or reduced enzyme activity and adverse reactions to ingested histamine, or that higher histamine levels are indeed present in these patients [135,136]. Therefore, HIT prevalence is unknown and there are no validated diagnostic methods for its diagnosis [135].

### 8.2. Clinical Characteristics

GI symptoms are the most common, including abdominal distension, postprandial fullness, diarrhea, abdominal pain, and constipation [149,150]. However, these symptoms lack specificity and can overlap with other GI disorders such as CD and DGBI [9]. Furthermore, histamine intolerance has been associated with additional symptoms beyond the gastrointestinal tract [149,150]. These include neurological manifestations like headaches and dizziness, cardiovascular symptoms such as tachycardia, hypotonia, and collapse, skin-related issues like itching, eczema, hives, swelling, and flushing, as well as respiratory symptoms including a runny nose, rhinitis, nasal congestion, and difficulty breathing [141,151,152]. These symptoms extend beyond the diagnosis of IBS [9,136]. The complex and variable nature of these symptoms poses a challenge in establishing a consensus regarding diagnostic criteria for histamine intolerance [141,151,152].

### 8.3. Diagnosis

The diagnosis of this condition remains challenging due to the non-specific nature of symptoms and the lack of validated diagnostic tools. Although diagnostic algorithms have been proposed, it is important to consider a broad range of potential causes of symptoms, including endogenous histamine release, and explore other underlying factors such as chronic urticaria, gastrointestinal diseases, mastocytosis, and allergic conditions [153]. Moreover, it is crucial to rule out the use of drugs that have been linked to an inhibition of the DAO enzyme when evaluating histamine intolerance [154] (Appendix A).

### 8.4. Treatment Approaches to Histamine Intolerance

At present, adhering to a low-histamine diet is the primary approach to managing symptoms associated with histamine intolerance [155,156]; however, due to the significant lack of evidence on this topic, a restrictive diet should be avoided. Anti-histamines have been proposed as adjunctive therapy in patients not responsive to diet [156]. However, there has been recent speculation about the potential of mast cell stabilizers and exogenous DAO supplementation as complementary therapy [157]. The purpose of this therapy is to improve the digestion of dietary histamine in individuals with histamine intolerance, who may have insufficient levels of this enzyme in their intestines [158]. However, there is currently a lack of long-term follow-up data on patients with histamine intolerance [153]. The main objective of the therapy is to prevent symptoms and resolve clinical manifestations associated with the condition [155].

#### 8.4.1. Dietary Approach

The primary suggested approach for preventing and managing histamine intolerance is to implement a diet that is either low in histamine or eliminates histamine from food intake [156]. However, the effectiveness of these diets lacks validation through randomized clinical trials. As previously mentioned, histamine is abundantly present in various food categories, and its concentrations can vary significantly due to multiple influencing factors [153]. Numerous clinical studies continuously provide increasing evidence of the efficacy of low-histamine diets in improving or alleviating symptoms [159,160,161,162,163,164,165,166]. Most of these studies have limitations in terms of the number of patients or duration of the dietary intervention [156]. However, there is a discrepancy among the various low-histamine diets regarding the list of foods to be excluded, leading to a lack of consensus on the dietary management of histamine intolerance [155]. Designing a low-histamine diet faces several challenges [156]. One of these challenges is the absence of agreement on the histamine level at which a food is considered low in histamine, with some sources suggesting a threshold of 1 mg/kg while others have a higher value [140]. Furthermore, some foods, including nuts, pineapple, and spinach, despite not containing high histamine levels, have been implicated in triggering the release of histamine from mast cells. The exact mechanism responsible for this potential effect remains unclear [167]. Additionally, some diets exclude foods containing putrescine and cadaverine due to reports suggesting that these biogenic amines might interfere with histamine degradation by the DAO enzyme at the intestinal level. Nevertheless, there is limited experimental evidence supporting this hypothesis [168]. As a result, the exclusion of these food categories from low-histamine diets exhibits a higher degree of variability [155].

#### 8.4.2. Antihistamines

No double-blind, placebo-controlled prospective studies have been conducted on the effectiveness of H1 and H2 receptor blockers in individuals experiencing adverse reactions to ingested histamine. Nevertheless, they have been used as second-line therapy in clinical practice due to their low rate of adverse events and the assumption, based on their mechanisms of action, that they could play a role in symptom alleviation. Specifically, there have been suggestions to utilize H1 blockers to alleviate flushing symptoms and H2 blockers to address nausea and vomiting [153].

#### 8.4.3. Mast Cell Stabilizers

Mast cell stabilizers function by stabilizing the mast cell membrane, thereby preventing the release of mediators like histamine [169]. These products are marketed as oral solutions to prevent food allergies and as topical solutions for the prevention of allergic rhinitis, asthma, or allergic conjunctivitis. Their optimal efficacy is achieved when administered before exposure to antigens [170]. In some cases, cromolyn, an oral mast cell stabilizer, has been prescribed at a dose of 100–200 mg, to be taken 20–30 min before meals [153]. However, there is insufficient evidence to support the use of mast cell stabilizers in patients with histamine intolerance.

#### 8.4.4. Oral Supplementation with Exogenous DAO

Similar to the current treatment for LI, the possibility of oral supplementation with exogenous DAO has been proposed by several authors to facilitate dietary histamine degradation and therefore to allow a less restrictive diet in terms of histamine content [171,172].

Currently, there are multiple alternatives on the market, mainly from porcine kidney extract but also DAO enzymes of plant origin [173]. Trials have been undertaken to investigate the efficacy of this treatment [151,157,158,174,175]. Despite variations in study design, enzyme dosage, intervention duration, and methods for measuring efficacy outcomes, the existing research consistently suggests that DAO supplements are effective in reducing the frequency and severity of symptoms [157]. In general, while the initial findings are promising, it is crucial to conduct more comprehensive clinical studies with robust experimental designs, longer treatment durations, and appropriately large sample sizes to establish the clinical effectiveness of this treatment [153].

## 9. Other Food-Specific Intolerances

Other more specific food intolerances have been generally reported, although evidence of their presence is lacking [7]. Indeed, despite the availability of numerous tests for diagnosing food intolerances, none of them have been validated, and most lack rigorous, blinded trials [176]. Consequently, a disparity exists between diagnosed food intolerances and individuals’ self-reported experiences, particularly in patients with IBS [9]. Self-diagnosed food intolerances frequently prompt the elimination of multiple foods from one’s diet, potentially resulting in nutritional deficiencies, psychological challenges, disruption of the gastrointestinal microbiota, and reduced quality of life due to limited dietary options and social activities [177]. Among these tests, confocal laser endomicroscopy has been of scientific interest since its introduction in the early 2000s. 

Confocal laser endomicroscopy (CLE) is an endoscopic imaging technique that provides real-time visualization of changes in the gut mucosa [178]. Using intravenous fluorescein allows high-resolution imaging at a microscopic level during ongoing endoscopy, akin to histological evaluation [179]. However, the clinical significance of CLE is not yet fully established [180]. In clinical endoscopy, two distinct CLE systems have been employed: probe-based CLE (pCLE) and endoscope-based CLE (eCLE) [180].

With the objective of both understanding the pathophysiology and finding a new diagnostic method, confocal laser endomicroscopy (CLE) was applied to IBS patients, food intolerance patients, food allergy, dyspepsia, and IBD patients [181]. The scientific rationale for applying this technique in food-intolerant individuals is that there may be reactions to some foods that trigger functional changes that go undetected by classic histology [181].

In a study conducted in 2014 by Annette Fritscher-Ravens et al., the researchers examined the role of confocal laser endomicroscopy (CLE) in individuals with self-reported food intolerances and IBS. They utilized CLE to observe the intestinal mucosa following food challenges in these patients. The study findings revealed that over half of the IBS patients experienced rapid onset reactions upon exposing their duodenal mucosa to the food antigens used during the food challenge, while no healthy control showed similar reactions. Five minutes after applying specific food suspensions to the duodenal mucosa of sensitive patients, confocal laser endomicroscopy revealed notable changes. These included an elevation in the number of intraepithelial lymphocytes, shedding of epithelial cells with subsequent formation of breaks leading to leaks, fluorescein secretion into the lumen, and edema accompanied by enlarged inter-villous spaces [182].

Bojarski et al. conducted a double-blinded prospective clinical study in non-coeliac disease patients suffering from IBS. Over two months, the researchers observed symptom improvement in 57% of patients with IBS who adhered to a GFD. Only 38 of these patients were correctly classified by CLE. Thus, the authors report poor specificity and sensitivity for endoscope-based eCLE for detecting NCWS, defined by the authors as a symptomatic improvement on a GFD [183].

In 2023, Gjini et al. conducted a novel study exploring the connection between functional abdominal pain and adverse food reactions using eCLE and local duodenal food challenges. The results showed that 67.6% of the patients responded to food challenges with fluorescein leakage into the duodenal lumen, and 23% exhibited spontaneous fluorescein leakage before the duodenal food challenge, which the authors interpreted as indicative of a leaky gut syndrome. No increase in IELs was noted. Overall, food exclusion therapy guided by the CLE results led to an improvement in 69.5% of the patients [184].

IgG blood testing is a test widely used in alternative medicine to identify the aliments provoking symptoms. Under typical conditions, when small quantities of food antigens enter the bloodstream, individuals in good health naturally generate and sustain increased IgG antibodies targeted toward these specific antigens. After consuming a meal, there are both antibodies and compounds created through the combination of food antigens with specific IgGs present in the bloodstream. It has been suggested that, in the context of Type III hypersensitivity reactions, IgG antibodies can form immune complexes with allergens found in food, leading to the initiation of mild inflammatory responses within the body [185]. Numerous studies have investigated their roles in both allergies and other disorders [186,187,188,189,190,191,192,193,194,195,196,197,198]. It was determined then that they do not play a causal role in eliciting food hypersensitivity reactions and do not provide information about food allergies. Scientific societies do not recommend their assessment as a test for food allergies [199].

Numerous costly alternative diagnostic methods for suspected food intolerances can be found online and are sometimes promoted to physicians or used by practitioners of complementary and alternative medicine [184]. The medical community does not widely accept these tests [200] since there is a lack of well-designed controlled trials confirming their efficacy [201]. One of these methods is the mediator release test (MRT), which measures the release of chemical mediators by white blood cells upon exposure to food antigens [202]. Another approach is ALCAT testing, which assesses immune cell reactions to a panel of food antigens [203]. Cytotoxic assays involve evaluating the effect of food antigens on white blood cells [204]. The electrodermal test measures skin conductance and purportedly detects changes associated with food intolerances. Hair analysis aims to identify food intolerances by analyzing mineral or heavy metal content in hair samples [205]. Iridology suggests that patterns in the iris reflect health conditions, including food intolerances [206]. Kinesiology involves muscle testing to identify food intolerances [207]. Bioresonance testing claims to assess energy frequencies related to food intolerances [205]. The pulse test suggests that changes in heart rate indicate food intolerance [208]. Sublingual or intradermal provocation–neutralization methods aim to desensitize individuals to food intolerances [209]. Lastly, ECIS^®^ (Electric Cell-substrate Impedance Sensing) is a scientific research tool used to study cell behavior and interactions, although its application for diagnosing food intolerances is limited and it is not commonly used in clinical practice for this purpose [205].

## 10. Conclusions

In recent years, an increasing number of people believe they have one or more food intolerances or allergies. In this review, we analyzed previous literature on food intolerances, examining and synthesizing existing studies. Our goal was to provide an overview of this field’s current state of knowledge, aiming to elucidate the nuances, identify gaps, and pinpoint potential areas for future research. Due to the nature of a narrative review, we did not conduct systematic research and the study selection is based on the authors’ knowledge and expertise in the field.

Non-celiac gluten sensitivity, lactose intolerance, and the rarer forms of genetic-based intolerances are more frequent nowadays due to the improved knowledge of the diseases and more accurate diagnostic testing. However, the ‘other’ food intolerances showed an even greater increase that may occur for several reasons. Some people may self-diagnose a food intolerance because they incorrectly attribute symptoms to foods they have eaten or from supportive but misleading health advice from family and friends or even just thanks to “Dr. Google”. However, another factor contributing to the increased frequency of diagnoses of other food intolerances is the widespread use of non-validated tests administered by both alternative medicine doctors and non-medical personnel. The reasons may be the poor response of currently available therapies for IBS, or the attractive messages easily found on the internet. As a result, the majority of ‘other food intolerant’ patients manage their diet by themselves, rather than seeking proper medical advice [210]. Many of the tests offered are not evidence-based, and often, the results lead to unwarranted self-imposed dietary restrictions that increase the risk of nutritional deficiency and affect patients’ social lives. Communicating clearly with our patients is essential, giving them the correct indications for restricting their diet only when necessary. Lactose intolerance and hereditary fructose intolerance have well-defined pathogenesis and have validated diagnostic tests; non-coeliac gluten sensitivity and FODMAP intolerance are mainly based on patients’ reports and lack tests to confirm their presence. Other reported intolerances, like non-hereditary fructose, sorbitol, and histamine intolerance still need more evidence and often cause unnecessary dietary restrictions. Most intolerance tests our patients perform lack scientific evidence of their validity. Therefore, the medical community should work to reduce their use since they often lead to problematic management of our patients. 

## Figures and Tables

**Figure 1 nutrients-15-04969-f001:**
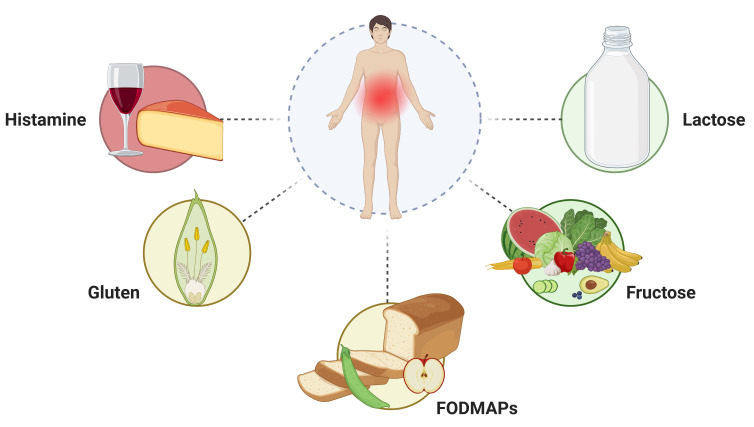
Main causes of reaction to specific food intake in intolerant individuals. Created with BioRender.com (accessed on 1 November 2023).

**Figure 2 nutrients-15-04969-f002:**
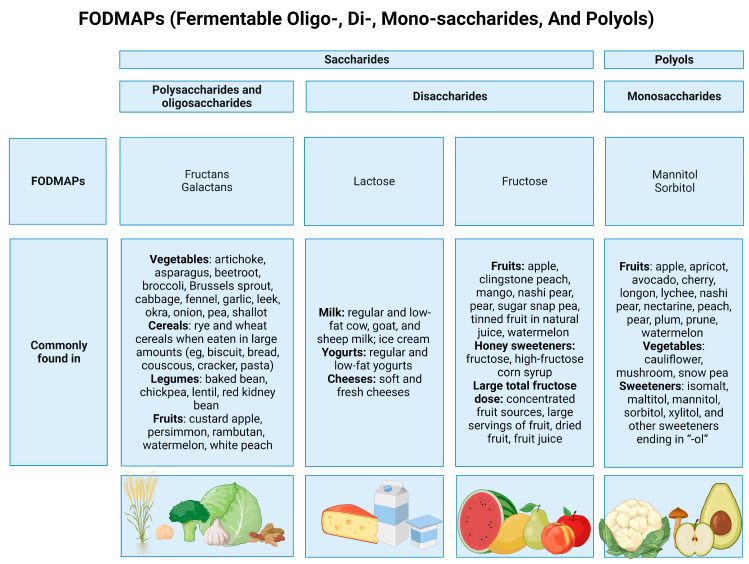
Food containing FODMAPs. Created with BioRender.com (accessed on 29 November 2023).

**Figure 3 nutrients-15-04969-f003:**
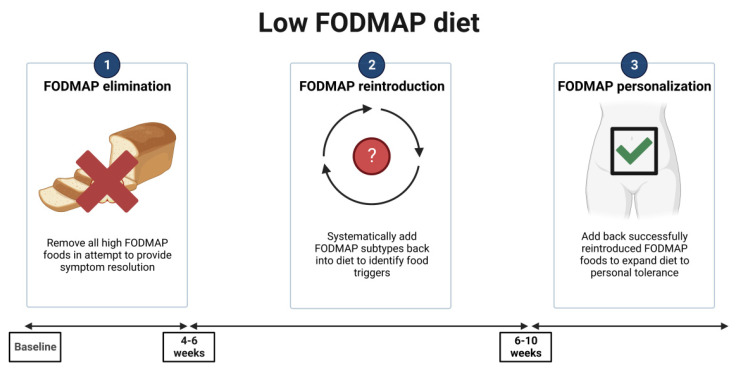
The top-down approach low-FODMAP diet. Created with Biorender.com. (accessed on 24 November 2023).

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
