# Peer review of "Myths and Facts about Food Intolerance: A Narrative Review"

_nutrients, 2023, doi:10.3390/nu15234969_

Round 1
Reviewer 1 Report
Comments and Suggestions for Authors
Please find comments as follows:
1-In abstract or introduction after mentioning “lower quality of life” Specify how these adverse reactions lead to dietary restrictions and a lower quality of life.
2-In abstract Consider using "common" or "widespread" food intolerances. as addressing all possible food intolerances is not possible.
3- What author means by giving the clinicians a guide on managing these cases: specify in the abstract if it is diagnostic, treatment, or general management.
4- It has been discussed in the abstract that this “often cause unneeded dietary restrictions: Provide examples or evidence supporting the claim that these intolerances often result in unnecessary dietary restrictions probably in the introduction you can discuss this.
5-Consider briefly discussing key findings from previous studies on food intolerances to Provide context and describe the unique aspects of your study and its novelty.
6- State the abbreviation IBS (Irritable Bowel Syndrome) upon first use for better understanding.
7- Figures are not clear, use higher quality
Comments on the Quality of English LanguageMinor
Reviewer 2 Report
Comments and Suggestions for Authors
Dear Authors,
Thank you for submitting the manuscript entitled “Myths and Facts about food intolerance: a narrative review“. The manuscript provides very important and useful information. It is well written and easy to read. However, some issues need to be clarified and here are some comments.
General comment
- there is no need to repeat the same reference at the end of every sentence referring to the same citation (e.g., lines 92-94; 102-104, 130-136; 404-411, etc.)
Specific comments:
2. Non-Coeliac Gluten/ Wheat Sensitivity – NCGWS
- for definition of gluten related disorders consider referring to Ludvigsson JF, et al. The Oslo definitions for coeliac disease and related terms. Gut. 2013 Jan;62(1):43-52.
2.1. Epidemiology and Diagnosis
- please cite the reference for diagnostic criteria for NCGS
- an important factor contributing to the misdiagnosis of NCGS is not only seronegative coeliac disease, but also not testing for celiac serology as it was not done in ref 14. Duodenal damage is not mentioned in the ref 14.
- the risk of undiagnosed CD (in those self-diagnosed) is not 42%, it ranges from 2% to 42%
2.3. Gluten or not Only Gluten That is the Question
- if it is true that “The majority of grains are gluten-containing foods” please cite the reference
2.4. Conclusions
- what does it mean – gluten challenge diet?
- line 172 – attention should be directed not towards GFD but towards individuals who are already on GFD (I suggest explaining why is that important)
- the last sentence “Additionally, there is a perception that a GFD is a healthy lifestyle, regardless of the side effects such as the higher fat content, the loss of dietary fiber, and deficiencies in dietary minerals and vitamins[38].” was not mentioned in the previous text, so I do not think it should be in the conclusion. However, it is an important point and if you chose to keep it needs be explained further.
3. FODMAPs intolerance
- please explain what FODMAP stands for (it is explained only in the abstract, not in the text)
- do you mean short-chain carbohydrates?
4. Lactose intolerance
4.3. Diagnosis
- line 389 - in separate portions?
5. Hereditary fructose intolerance
- please redefine the definition of HFI and add what causes the GI symptoms and potential long-term organ damage – ingestion of fructose or HFI can result in…
5.2. Diagnosis
- there is no need to explain basic autosomal recessive inheritance
- line 505 – “highly specific sensitive“ meaning highly specific and sensitive?
6. Non-hereditary fructose intolerance
- consider commenting Broekaert, IJ; et al. An ESPGHAN Position Paper on the Use of Breath Testing in Paediatric Gastroenterology. Journal of Pediatric Gastroenterology and Nutrition 74(1):p 123-137,
10. Conclusions
- celiac disease is autoimmune disease, not intolerance
References:
- 24 references older than 20 years
- References 19, 92, 200 are not correct – authors and journal are lacking
- Check the reference 197, 198, 204
Reviewer 3 Report
Comments and Suggestions for Authors
Dear authors and editor,
The manuscript titled "Myths and Facts about food intolerance: a narrative review" aimed to clarify the myths and reality of any food intolerance, giving the clinicians a guide on managing these cases.
There are many minor issues I'd like the authors resolve.
Abstract
1-Change the keywords. Delete the words "non-coeliac gluten sensitivity", "histamine intolerance ", "Hereditary fructose intolerance" and "Congenital sucrase-isomaltase deficiency ". Not found in the MeSH (Medical Subject Headings).Change to FODMAP Diet, Fructose Intolerance,Sucrase-isomaltase deficiency, congenital....
2-The title is appropriate. The authors identify the study design.
Introduction
3-Adequate: The most important concepts of the subject to be developed are identified.The authors have done a great job in describing the findings.
Materials and Methods
- 4-It is recommended to include a brief section on methodology. What criteria did you use to conduct the narrative review? Where did you look to support your conclusions? ...
- Discussion
- 5-A discussion section indicating the limitations is recommended.
Conclusion
- 6-Adequate:The objectives are answered in the conclusions.
Reference
- 7-adequate: Complies with the journal's standards.
I recommend the authors to review the article "Writing narrative style literature reviews" https://www.researchgate.net/publication/288039333_Writing_narrative_style_literature_reviews
Round 2
Reviewer 3 Report
Comments and Suggestions for Authors
Thanks to the authors for their responses.